# Surface Approximation by Means of Gaussian Process Latent Variable Models and Line Element Geometry

**Ivan De Boi** [1,*], **Carl Henrik Ek** [2] **and Rudi Penne** [1]

1 Research Group InViLab, Department Electromechanics, Faculty of Applied Engineering, University of Antwerp, B 2020 Antwerp, Belgium
2 Department of Computer Science and Technology University of Cambridge, Cambridge CB3 0FD, UK
* Correspondence: ivan.deboi@uantwerpen.be

**Abstract:** The close relation between spatial kinematics and line geometry has been proven to be fruitful in surface detection and reconstruction. However, methods based on this approach are limited to simple geometric shapes that can be formulated as a linear subspace of line or line element space. The core of this approach is a principal component formulation to find a best-fit approximant to a possibly noisy or impartial surface given as an unordered set of points or point cloud. We expand on this by introducing the Gaussian process latent variable model, a probabilistic non-linear non-parametric dimensionality reduction approach following the Bayesian paradigm. This allows us to find structure in a lower dimensional latent space for the surfaces of interest. We show how this can be applied in surface approximation and unsupervised segmentation to the surfaces mentioned above and demonstrate its benefits on surfaces that deviate from these. Experiments are conducted on synthetic and real-world objects.

**Keywords:** surface approximation; surface segmentation; surface denoising; gaussian process latent variable model; line geometry; line elements

**MSC:** 60G15

## 1. Introduction

Extracting structural information as shapes or surfaces from an unordered set of 3D coordinates (point cloud) has been an important topic in computer vision [1]. It is a crucial part of many applications such as autonomous driving [2], scene understanding [3], reverse engineering of geometric models [4], quality control [5], simultaneous localization and mapping (SLAM) [6] and matching point clouds to CAD models [7]. Over the last decade, hardware developments have made the acquisition of those point clouds more affordable. As the availability, ease of use and hence the popularity of various 3D sensors increases so does the need for methods to interpret the data they generate.

However, in this work, we mainly focus on detecting simple geometrical surfaces such as planes, spheres, cylinders, cones, spiral and helical surfaces, surfaces of revolution, etc. as described in [8]. Examples of these surfaces can be found in Figure 1. In [8], the close relation between these shapes, spatial kinematics and line geometry are formulated. A point cloud, as a set of noisy points on a surface, is transformed into a set of normals (also referred to as normal lines or normal vectors) that show exploitable properties for that surface. For instance, the normals of a sphere intersect in a single point, the normals of a surface of revolution intersect in an axis of rotation, and the normals of a helical surface can be seen as path normals of a helical motion. These insights led to applications in surface reconstruction and robotics [8,9]. Later, their method was refined in [10,11] to address pipe surfaces, profile surfaces and developable surfaces in general. In [12], the authors introduced principal component analysis (PCA) to approximate the set of normals. This laid the groundwork for a more general approach in [13] using so-called *line elements*.

These are constructed for every point of the point cloud. They are formed by the (Plücker) coordinates of the normal line and the surface point which lies on that line. The key insight of their work is that the line elements of simple geometric surfaces lie on a linear subspace in $\mathbb{R}^7$, which can be found by solving an ordinary eigenvalue problem. We elaborate more on this approach in Section 2.2.

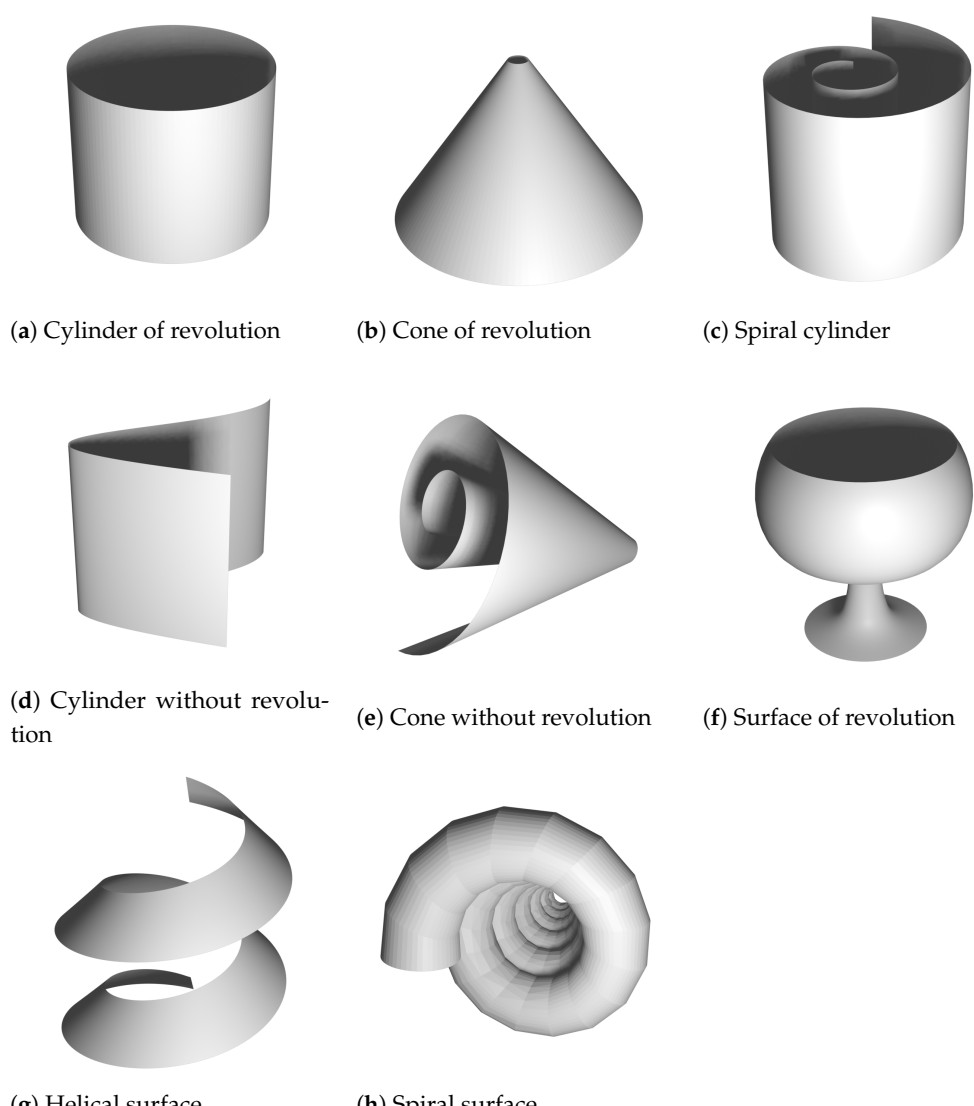

(**a**) Cylinder of revolution

(**b**) Cone of revolution

(**c**) Spiral cylinder

(**d**) Cylinder without revolution

(**e**) Cone without revolution

(**f**) Surface of revolution

(**g**) Helical surface

(**h**) Spiral surface

**Figure 1.** Examples of equiform kinematic surfaces.

Although a mathematically very elegant way to describe 3D surfaces, this approach does have several drawbacks. First, the surface classification is strict. This means that only very primitive shapes can be treated. Real-world objects do not always fall neatly into one of these categories. For example, imperfect shapes like a slight bend plane, a sphere with a dent or shapes found in nature or human anatomy. Blindly following this method, results in a misrepresentation of the data. Second, because PCA minimises an L2-norm, it is very sensitive to outliers. This can be mitigated by iterative RANSAC or by downweighting the outliers. However, this comes at the cost of increased computation time. Third, real-world point clouds show various other imperfections like non-uniform sampling, noise, missing regions, etc. This highly affects the quality of the representation. Fourth, the authors of [10–13] propose to look for *small* eigenvalues. The obvious question arises: when is an eigenvalue small? Even though some guidelines are provided, these threshold values remain domain specific and are challenging to set.

Most of these drawbacks can be attributed to the eigenvalue problem (or its PCA formulation) used to find an appropriate linear subspace in $\mathbb{R}^7$. In essence, this is a linear dimensionality reduction from the seven-dimensional space of line elements to a lower-dimensional latent space. In this work, we build on that method by introducing the Gaussian process latent variable model (GPLVM) [14] as an alternative. This allows for a non-linear relationship between a latent space and a higher dimensional data space, where observations are made. We implement a multi-output Gaussian process (seven outputs in this case) and try to find a mapping from a lower dimensional latent space to the line elements. Several variants on this theme exist, which we explain in more depth in Section 2.5. No longer confined by the linearity of PCA, our models can handle a wider range of shapes. Moreover, Gaussian processes handle noise very well, even in low data regimes [15]. The GPLVM places a Gaussian process prior to the mapping from the latent to the observed space. By exploiting strong priors, we can significantly reduce the amount of data needed for equally accurate predictions.

In our approach, we can handle shapes (or surfaces) that fall in the categories described in [8–13] but also shapes that significantly deviate from these. For instance, a surface of revolution whose central axis is not a straight line or an imperfect plane with one of the corners slightly bent. In fact, we drop the strict classification and allow for shapes that can be seen as somewhere in between the categories. This makes our methods more appropriate for handling surfaces that can be found in nature or when modelling the human body. Moreover, our formulation can handle multiple types of subsurfaces at once. This means we can perform segmentation in the latent space.

For completeness, we mention that in recent years various deep learning techniques have been successfully introduced to discover objects in point clouds. A thorough overview can be found in [16] and more recently in [1,17]. These techniques vary from ordinary multilayer perception models (MLP) to convolutional- and graph-based models. Numerous datasets have been made public to encourage the field further to develop new models (e.g., ScanObjectNN [18], ShapeNet [19], ScanNet [20], KITTI Vision Benchmark Suite [21], ModelNet40 [22,23], . . . ). Generally, these data-driven models are trained on more complex shapes: vehicles, urban objects, and furniture, . . . These models are specifically designed for detecting obstacles, such as vehicles in autonomous driving, but not so for accurate object reconstruction from detailed 3D scanning. In this work, we focus on the latter. Moreover, whenever a new shape has to be learned, the underlying model has to be trained again.

To summarise, for every point on a given point cloud, we can formulate a so-called line element. Dimensionality reduction on the set of line elements reveals the characteristics of the surface captured by that point cloud. Existing methods rely on PCA, which is a linear mapping. In contrast, our model is built on the Gaussian process latent variable model, which allows for non-linear mapping. This results in a more nuanced way of representing the surface. The main contributions of this work are the following:

- We expand existing methods based on kinematic surfaces and line element geometry by introducing GPLVM to describe surfaces in a non-linear way.
- We apply our method to surface approximation.
- We test our method to perform unsupervised surface segmentation.
- We demonstrate our method to perform surface denoising.

All of our 3D models, sets of line elements, trained GPLVM, notebooks with code and many more experiments and plots can be found on our GitHub repository (https://github.com/IvanDeBoi/Surface-Approximation-GPLVM-Line-Geometry, accessed on 10 January 2023).

The rest of this paper is structured as follows. In the next section, we give some theoretical background on line element geometry, kinematic surfaces, approximating the data, Gaussian processes and Gaussian process latent variable models in particular. The third section describes the results of our method applied to surface approximation, surface segmentation and surface denoising. Section four presents a discussion of our findings.

## 2. Materials and Methods

### 2.1. Line Element Geometry

In projective 3-space $\mathbb{P}^3$, a straight line **L** can be represented by a direction vector **l** and an arbitrary point $x$ on that line. The so-called moment vector $\bar{\mathbf{l}}$ for that line with respect to the origin, can be written as

$$\bar{\mathbf{l}} = \mathbf{x} \times \mathbf{l}, \tag{1}$$

where **x** are the coordinates of $x$ in $\mathbb{R}^3$. The *Plücker coordinates* for a line are defined as $(\mathbf{l}, \bar{\mathbf{l}}) = (l_1 : l_2 : l_3 : l_4 : l_5 : l_6)$ [24]. These are homogeneous coordinates, meaning they are scale invariant. Notice that the scale factor is determined by the norm of the direction vector. Moreover, they are independent of the choice of $x$. Since we are not concerned about the orientation, $(\mathbf{l}, \bar{\mathbf{l}})$ and $(-\mathbf{l}, -\bar{\mathbf{l}})$ describe the same line, which also follows from the homogeneity.

For example, a line **L** is spanned by two given points $x$ and $y$, possibly at infinity. By following the notation in [24], we write the homogenous coordinates for $x$ and $y$ as $(x_0, \mathbf{x})$ and $(y_0, \mathbf{y})$ respectively. Then, the homogenous Plücker coordinates for **L** are found as

$$\mathbf{L} := (\mathbf{l}, \bar{\mathbf{l}}) = (x_0 \mathbf{y} - y_0 \mathbf{x}, \mathbf{x} \times \mathbf{y}) \in \mathbb{R}^6. \tag{2}$$

Not every combination of six numbers yields a straight line. To do so, the following condition is necessary and sufficient:

$$\mathbf{l} \cdot \bar{\mathbf{l}} = 0. \tag{3}$$

This is called the *Grassmann–Plücker relation*. Plücker coordinates can also be regarded as homogeneous points coordinates in projective 5-space $\mathbb{P}^5$, where straight lines are points lying on a quadric given by the equation

$$l_1 l_4 + l_2 l_5 + l_3 l_6 = 0. \tag{4}$$

This quadric is called the *Klein quadric* and is denoted as $M_2^4$. The interpretation of points in projective 5-space has proved useful in a variety of line geometry applications [24].

Plücker coordinates of a line in $\mathbb{R}^3$ can be extended to *line elements* by adding a specific point $x$ on that line [13]. To do so, we start by choosing an orientation for the unit direction vector **l** of the line. A seventh coordinate $\lambda$ is needed to locate $x$ on that line, which can be defined as

$$\lambda = \mathbf{x} \cdot \mathbf{l}. \tag{5}$$

Notice that the norm of **l** matters, which is why we work with the (normalised) unit direction vector. This yields the seven-tuple $(\mathbf{l}, \bar{\mathbf{l}}, \lambda)$ of coordinates for a line element based on a line and a point, in which $\|\mathbf{l}\| = 1$ and $\mathbf{l} \cdot \bar{\mathbf{l}} = 0$.

Each point on a smooth surface $\Phi$ of a 3D volume has an outward unit normal vector **n**. For every point $x$ on that surface, a line element can be defined as $(\mathbf{n}, \mathbf{x} \times \mathbf{n}, \mathbf{x} \cdot \mathbf{n})$. These line elements constitute an associated surface $\Gamma(\Phi)$ in $\mathbb{R}^7$. An important property of many simple geometrical shapes in $\mathbb{R}^3$ (planes, spheres, cones, . . . ), is that their $\Gamma(\Phi)$ is contained in a linear subspace of $\mathbb{R}^7$. We will see in Section 2.2 that this aspect can be exploited in surface approximation, surface segmentation and surface denoising.

### 2.2. Kinematic Surfaces

Rigid body motions can be seen as a superposition of rotations and translations. These can be extended by adding a scaling, making them the family of *equiform motions,* also known as *similarities* [10]. Such a one-parameter motion $M(t)$ is either a rotation, translation, a central similarity, a spiral motion or a combination of any of them. The velocity vector field of $M(t)$ is constant (time-independent) and can be written as

$$\mathbf{v}(\mathbf{x}) = \bar{\mathbf{c}} + \gamma \mathbf{x} + \mathbf{c} \times \mathbf{x}, \tag{6}$$

where $\bar{\mathbf{c}}$, $\gamma \mathbf{x}$ and $\mathbf{c} \times \mathbf{x}$ are the translation, scale and rotation component of the velocity vector $\mathbf{v}$ at a point $x$. A curve undergoing an equiform motion forms an *equiform kinematic surface*.

As defined in [13], a *linear complex of line elements* is the set of line elements whose coordinates $(\mathbf{l}, \bar{\mathbf{l}}, \lambda)$ satisfy the linear equation

$$\bar{\mathbf{c}} \cdot \mathbf{l} + \mathbf{c} \cdot \bar{\mathbf{l}} + \gamma \lambda = 0, \tag{7}$$

where $(\mathbf{c}, \bar{\mathbf{c}}, \gamma)$ is de coordinate vector of the complex. The following theorem from [25] shows the relation between linear complexes of lines and equiform kinematics:

**Theorem 1.** *The surface normal elements of a regular $C^1$ surface in $\mathbb{R}^3$ are contained in a linear line element complex with coordinates $(\mathbf{c}, \bar{\mathbf{c}}, \gamma)$ if and only if the surface is part of an equiform kinematic surface. In that case, the uniform equiform motion has the velocity vector field as given in Equation (6).*

Here, we will give an overview of such motions $M(t)$ and their corresponding surfaces $\Phi$. For a thorough explanation of these (and multiple applications), we refer the reader to the works [8,10–13,25,26].

- $\gamma = 0$:
    - $\mathbf{c} = 0, \bar{\mathbf{c}} = 0$: $M(t)$ is the identical motion (no motion at all).
    - $\mathbf{c} = 0, \bar{\mathbf{c}} \neq 0$: $M(t)$ is a translation along $\bar{\mathbf{c}}$ and $\Phi$ is a cylinder (not necessarily of revolution).
    - $\mathbf{c} \neq 0, \mathbf{c} \cdot \bar{\mathbf{c}} = 0$: $M(t)$ is a rotation about an axis parallel to $\mathbf{c}$ and $\Phi$ is a surface of revolution.
    - $\mathbf{c} \neq 0, \mathbf{c} \cdot \bar{\mathbf{c}} \neq 0$: $M(t)$ is a helical motion about an axis parallel to $\mathbf{c}$ and $\Phi$ is a helical surface.

- $\gamma \neq 0$:
    - $\mathbf{c} \neq 0$: $M(t)$ is a spiral motion and $\Phi$ is a spiral surface.
    - $\mathbf{c} = 0$: $M(t)$ is a central similarity, and $\Phi$ is a conical surface (not necessarily of revolution).

Examples of these surfaces can be found in Figure 1.

This alternative way of describing surfaces as linear complexes of line elements opens up a new way of studying them, as explained below.

*2.3. Approximating the Complex*

Suppose a scanning process results in a set of points $X$ (a point cloud), i.e., the results of the scanning process. The aim is to determine the type of surface $\Phi$ on which these points lie. This knowledge would allow us to reconstruct the surface using its underlying geometrical properties. For instance, if we know our points result from the scan of a surface of revolution, we could determine the central axis etc. So, we are interested in the (linear) complex (of line elements) that best describes the given points. Its coordinates $(\mathbf{c}, \bar{\mathbf{c}}, \gamma)$ determine the type of surface [13].

First, we calculate the unit normal vectors from the point cloud at every point. This topic has been very well documented in the literature. We refer the reader to [27] for a more in-depth discussion. For each $x_i$ in $X$ with $i = 1, 2, \dots, N$ we obtain a unit normal vector $\mathbf{n_i}$. From these normal vectors and corresponding points, we calculate their line elements $(\mathbf{n_i}, \mathbf{x_i} \times \mathbf{n_i}, \mathbf{x_i} \cdot \mathbf{n_i})$.

Second, according to Equation (7), a complex with coordinates $(\mathbf{c}, \bar{\mathbf{c}}, \gamma)$ that best fits these line elements minimises

$$F(\mathbf{c}, \bar{\mathbf{c}}, \gamma) = \sum_{i=1}^{N} (\bar{\mathbf{c}} \cdot \mathbf{l_i} + \mathbf{c} \cdot \bar{\mathbf{l}}_\mathbf{i} + \gamma \lambda_i)^2, \tag{8}$$

under the condition $\mathbf{c}^2 + \bar{\mathbf{c}}^2 + \gamma^2 = 1$. We follow the notation used in [12], in which $\mathbf{a}^2 = \mathbf{a} \cdot \mathbf{a}$. For this condition to make sense, we normalise our point cloud such that $\max\|x_i\| \approx 1$. We also centre it around the origin. We can rewrite this as

$$F(\mathbf{c}, \bar{\mathbf{c}}, \gamma) = (\mathbf{c}, \bar{\mathbf{c}}, \gamma) M (\mathbf{c}, \bar{\mathbf{c}}, \gamma)^T, \tag{9}$$

where $M = \sum_{i=1}^{N} (\bar{\mathbf{l}}_\mathbf{i}, \mathbf{l}_\mathbf{i}, \lambda_i)^T (\bar{\mathbf{l}}_\mathbf{i}, \mathbf{l}_\mathbf{i}, \lambda_i)$. This is an ordinary eigenvalue problem. The smallest eigenvalue of $M$ corresponds to an eigenvector $(\hat{\mathbf{c}}, \hat{\bar{\mathbf{c}}}, \hat{\gamma})$ which best approximates Equation (7) for the given $(\mathbf{l}_\mathbf{i}, \bar{\mathbf{l}}_\mathbf{i}, \lambda_i)$.

Some surfaces are invariant under more than one one-parameter transformation [8,10–13,25,26]. In that case, $k$ small eigenvalues appear as solutions to Equation (8). The corresponding eigenvectors can be seen as a basis for a subspace in $\mathbb{R}^7$. We list the possibilities below:

- $k = 4$: Only a plane is invariant to four independent uniform motions.
- $k = 3$: A sphere is invariant to three independent uniform motions (all rotations).
- $k = 2$: The surface is either a cylinder of revolution, a cone of revolution or a spiral cylinder.
- $k = 1$: The surface is either a cylinder without revolution (pure translation), a cone without revolution (central similarity), a surface of revolution, a helical surface or a spiral surface.

Further examination of the coordinate $(\mathbf{c}, \bar{\mathbf{c}}, \gamma)$ determines the exact type of surface, as described in Section 2.2. Multiple examples, applications and variations on this theme can be found in above mentioned references.

Although this is a very elegant and powerful approach, some issues are discussed in the works listed above. First, this method is very sensitive to outliers. A solution proposed by the authors is to apply a RANSAC variant or to downweigh the outliers by iteratively

$$M = \frac{1}{\sum \sigma_i} \sum_{i=1}^{N} \sigma_i (\bar{\mathbf{l}}_\mathbf{i}, \mathbf{l}_\mathbf{i}, \lambda_i)^T (\bar{\mathbf{l}}_\mathbf{i}, \mathbf{l}_\mathbf{i}, \lambda_i). \tag{10}$$

This obviously results in longer computation times. Second, numerical issues can arise calculating the eigenvalues (especially for planes and spheres). Third, some shapes do not fall into the classification of these simple geometric forms. This is certainly the case for organic surfaces that can be found in nature or when modelling the human body. Reconstructing a surface based on a simple geometric shape is obviously only valid if the surface resembles the shape well. Fourth, some shapes are either a combination or a composition of the elementary simple shapes (e.g., pipe constructions). In this case, the question arises of what constitutes as a small eigenvalue and where to draw the line between steadily increasing values. Even though some guidance is given in the literature, these thresholds are often application-specific parameters.

Our approach provides a solution for these issues, by finding a representative lower dimensional latent space for the line elements in a more flexible non-linear way. This is no longer a linear subspace in $\mathbb{R}^7$.

### 2.4. Gaussian Processes

By definition, a Gaussian process (GP) is a stochastic process (a collection of random variables), with the property that any finite subset of its variables is distributed as a multivariate Gaussian distribution. It is a generalization of the multivariate Gaussian distribution to infinitely many variables. Here, we only give an overview of the main aspects. We refer the reader to the book [15] for a more detailed treatise.

Let a dataset $\mathcal{D} = \{X, \mathbf{y}\}$ consist of $n$ observations, where $X = [\mathbf{x}_1, \mathbf{x}_2, \dots, \mathbf{x}_n]^T$ is an $n \times d$ matrix of $n$ input vectors of dimension $d$ and $\mathbf{y} = [y_1, y_2, \dots, y_n]^T$ is a vector of continuous-valued scalar outputs. These data points are also called training points. In regression, the aim is to find a mapping $f : \mathbb{R}^d \to \mathbb{R}$,

$$y = f(\mathbf{x}) + \epsilon, \quad \epsilon \sim \mathcal{N}(0, \sigma_\epsilon^2), \tag{11}$$

with $\epsilon$ being identically distributed observation noise. In this work, this mapping is implemented by a Gaussian process. As stated above, a Gaussian process generalises the multivariate Gaussian distribution to infinitely many variables. Just like the multivariate Gaussian distribution is fully defined by its mean vector and covariance matrix, a Gaussian process is fully defined by its mean as a function $m(\mathbf{x})$ and *covariance function $k(\mathbf{x}, \mathbf{x}')$*. It is generally denoted as $f(\mathbf{x}) \sim \mathcal{GP}(m(\mathbf{x}), k(\mathbf{x}, \mathbf{x}'))$. The covariance function is parametrised by a vector of hyperparameters $\boldsymbol{\theta}$. By definition, a GP yields a distribution over a collection of functions that have a joint normal distribution [15],

$$\begin{bmatrix} \mathbf{f} \\ \mathbf{f}_* \end{bmatrix} \sim \mathcal{N}\left( \begin{bmatrix} \boldsymbol{m}_X \\ \boldsymbol{m}_{X_*} \end{bmatrix}, \begin{bmatrix} \mathbf{K}_{X,X} & \mathbf{K}_{X,X_*} \\ \mathbf{K}_{X_*,X} & \mathbf{K}_{X_*,X_*} \end{bmatrix} \right), \tag{12}$$

where $X$ are the input vectors of the $n$ observed training points and $X_*$ are the $n_*$ input vectors of the unobserved test points. The mean value for $\mathbf{X}$ is given by $\boldsymbol{m}_X$. Likewise, the mean value for $X_*$ is given by $\boldsymbol{m}_{X_*}$. The covariance matrices $\mathbf{K}_{X,X}$, $\mathbf{K}_{X_*,X_*}$, $\mathbf{K}_{X_*,X}$ and $\mathbf{K}_{X,X_*}$ are constructed by evaluating the covariance function $k$ at their respective pairs of points. In real-world applications, we do not have access to the latent function values. We are depending on noisy observations $\mathbf{y}$.

The conditional predictive posterior distribution of the GP can be written as:

$$\mathbf{f}_* | X, X_*, \mathbf{y}, \boldsymbol{\theta}, \sigma_\epsilon^2 \sim \mathcal{N}(\mathbb{E}(\mathbf{f}_*), \mathbb{V}(\mathbf{f}_*)), \tag{13}$$

$$\mathbb{E}(\mathbf{f}_*) = \boldsymbol{m}_{X_*} + \mathbf{K}_{X_*,X} \left[ \mathbf{K}_{X,X} + \sigma_\epsilon^2 I \right]^{-1} \mathbf{f}, \tag{14}$$

$$\mathbb{V}(\mathbf{f}_*) = \mathbf{K}_{X_*,X_*} - \mathbf{K}_{X_*,X} \left[ \mathbf{K}_{X,X} + \sigma_\epsilon^2 I \right]^{-1} \mathbf{K}_{X,X_*}. \tag{15}$$

The hyperparameters $\boldsymbol{\theta}$ are usually learned by using a gradient-based optimisation algorithm to maximise the log marginal likelihood,

$$\log p(\mathbf{y}|\theta, X) \propto -\frac{1}{2} \left[ \mathbf{y}^T \left[ \mathbf{K}_{X,X} + \sigma_\epsilon^2 I \right] \mathbf{y} + \log |\mathbf{K}_{X,X} + \sigma_\epsilon^2 I)| \right], \tag{16}$$

which is a combination of a data fit term and complexity penalty and, thus, automatically incorporates Occam's Razor [15]. This guards the Gaussian process model against overfitting. In our experiments, we use BFGS, a quasi-Newton method described in [28]. The linear system $\left[ \mathbf{K}_{X,X} + \sigma_\epsilon^2 I \right] \mathbf{y}$ is often calculated by first calculating the Cholesky decomposition factor $L$ of $\left[ \mathbf{K}_{X,X} + \sigma_\epsilon^2 I \right]$ and then solving

$$\left[ \mathbf{K}_{X,X} + \sigma_\epsilon^2 I \right] \mathbf{y} = L^T \setminus (L \setminus \mathbf{y}). \tag{17}$$

In the literature, many kernel functions have been extensively studied and reviewed. An overview can be found in [29]. A very popular kernel is the squared exponential kernel. It is suited for a wide range of applications because it is infinitely differentiable and, thus, yields smooth functions. Moreover, it only has two tunable hyperparameters. It has the form:

$$k_{SE}(\mathbf{x}, \mathbf{x}') = \sigma_f^2 \exp\left( -\frac{|\mathbf{x} - \mathbf{x}'|^2}{2l^2} \right), \tag{18}$$

in which $\sigma_f^2$ is a scale factor and $l$ is the length-scale that controls the decline of the influence of the training points with distance. For the squared exponential kernel the hyperparameters $\boldsymbol{\theta}_{SE}$ are $\left\{ \sigma_f^2, l \right\}$. For the function $k : X \times X \to \mathbb{R}$ to be a valid kernel, it

must be positive semi-definite (PSD), which means that for any vector $\mathbf{x} \in X^d$, the kernel matrix $K$ is positive semi-definite. This implies that $\mathbf{x}^T K \mathbf{x} \geq 0$ for all $\mathbf{x} \in \mathbb{R}^d$.

In this work, we do implement a different length-scale parameter for every input dimension. This technique is called automatic relevance determination (ARD) and allows for functions that vary differently in each input dimension [29]. The kernel has the form:

$$k_{SEARD}(\mathbf{x}, \mathbf{x}') = \sigma_f^2 \exp\left( -\frac{1}{2} \sum_{j=1}^{d} \left( \frac{|\mathbf{x}_j - \mathbf{x}_j'|}{l_j} \right)^2 \right), \tag{19}$$

in which $l_j$ is a separate length-scale parameter for each of the $d$ input dimensions.

### 2.5. Gaussian Process Latent Variable Models

Principal component analysis (PCA) transforms a set of data points to a new coordinate system, in which the greatest variance is explained by the first coordinate (called the first principal component), the second greatest variance by the second coordinate, etc. This reprojection of the data can be exploited in dimensionality reduction by dropping the components with the smallest variance associated. The result will still contain most of the information of the original data. This method can also be explained as a statistical model known as probabilistic PCA (PPCA) [30], which implies that the principal components associated with the largest variance also maximise the likelihood of the data.

In dimensionality-reduction, the representation of the original data by its retained principal components can be interpreted as a latent variable model, in which the $n$ latent variables $\mathbf{X} = \begin{bmatrix} \mathbf{x}_1, \mathbf{x}_2, \ldots, \mathbf{x}_n \end{bmatrix}^T$ are of a dimension $k$ that is lower than the dimension $d$ of the original data. PPCA requires a marginalisation of those latent variables and an optimisation of the mapping from the latent space to the data (observation) space. For $n$ $d$-dimensional observations $\mathbf{Y} = \begin{bmatrix} \mathbf{y}_1, \mathbf{y}_2, \ldots, \mathbf{y}_n \end{bmatrix}^T$ we can write

$$\mathbf{y}_i = \mathbf{W}\mathbf{x}_i + \epsilon, \quad \epsilon \sim \mathcal{N}(0, \sigma_\epsilon^2), \tag{20}$$

in which $\mathbf{x}_i$ is a $k$-dimensional latent variable with $k < d$, $\mathbf{W}$ is a $d \times k$ matrix representing the mapping and $\epsilon$ is observation noise.

In [14,31], a dual approach is proposed by marginalising the mapping $\mathbf{W}$ and optimising the latent variables $\mathbf{X}$. This approach is called the Gaussian process latent variable model (GPLVM) and is achieved by maximising the Gaussian process likelihood $\mathcal{L}$ with respect to the latent variables. We optimise

$$\mathcal{L}(\mathbf{X}) = -\frac{dn}{2} \log 2\pi - \frac{d}{2} \log |\mathbf{K}| - \frac{1}{2} \mathrm{tr}(\mathbf{K}^{-1} \mathbf{Y} \mathbf{Y}^\top), \tag{21}$$

with respect to $\mathbf{X}$. It is proved in [14] that this approach is equivalent to PCA when using a linear kernel to compose $\mathbf{K}$, which can be written as

$$k_{linear}(\mathbf{x}, \mathbf{x}') = \mathbf{x}^T \mathbf{x}'. \tag{22}$$

However, by choosing a nonlinear kernel, we can establish a nonlinear relationship between the latent variables $\mathbf{X}$ and the observations $\mathbf{Y}$. This relationship can also be seen as placing a Gaussian process prior on each column of the $n \times d$ matrix $\mathbf{Y}$ and allows for a more flexible mapping between latent and data space.

In the original GPLVM, the unobserved inputs are treated as latent variables which are optimised. Another approach is to variationally integrate them and compute a lower bound on the exact marginal likelihood of the non-linear variable model. This is known as the Bayesian Gaussian process latent variable model (BGPLVM) [32]. It is more robust to overfitting and can automatically detect the relevant dimensions in the latent space. These dimensions are characterised by a larger Automatic Relevance Determination (ARD)

contribution, which is the inverse of the length-scales $l_j$ in Equation (19). Every component in data space is a vector whose components are formed by as many Gaussian processes as there are input dimensions. These ARD contributions determine the weight of the outcomes of each of the Gaussian processes and, thus, its corresponding input dimension. Less relevant dimensions result in longer length scales and are pruned out. In this work, we exploit this by performing this Bayesian non-linear dimensionality reduction on the seven-dimensional line elements. For most shapes in 3D, a lower-dimensional representation in a latent space can be found, as we show below.

The GPLVM is a map from latent to data space. As such, it preserves local distances between points in the latent space. However, this does not imply that points close by in the data space are also close by in the latent space. To incorporate this extra feature, one can equip the GPLVM with a so-called back constraint [33]. This constraint is accomplished by introducing a second distance-preserving map from the data to the latent space. We refer to this model as the back-constrained Gaussian process latent variable model (BCGPLVM). A thorough review of GPLVM and its variants, including more than the ones mentioned here, can also be found in [34,35] and more recently in [36,37].

### 2.6. Our Approach

In this section, we explain how all of the above-mentioned concepts come together in our approach. To recap, we can represent a straight line in 3D space by Plücker coordinates, which are six-tuples. By adding a seventh component, we can specify a point on that line. We can do this for every $n$ point in a given point cloud. The line we choose through each point is the normal line to the surface that is captured by that point cloud. We, thus, obtain a set of seven-dimensional line elements, that captures the information about the surface we want to examine.

The theory of kinematic surfaces links the line elements that are contained in a linear line element complex to an equiform kinematic surface. Finding this complex comes down to solving an ordinary eigenvalue problem. The dimensionality of the linear subspace, in which the line elements live, and the resulting eigenvalues determine the type of surface. In essence, this is dimensionality reduction on the seven-dimensional line elements via PCA.

However, PCA is a linear mapping. In contrast, our model is built on the Gaussian process latent variable model (GPLVM), which allows for non-linear mapping. This results in a more nuanced way of representing the surface. Our model is only given the seven-dimensional line elements and finds the mapping from a latent (unobserved) space to these line elements. Each of the seven dimensions of the line elements is assigned to a Gaussian process (GP). The outputs (predictions) of those GPs are the components of the line elements. The inputs (training data) are the latent points, which are treated as variables of the overall model and are optimised (or integrated out in the Bayesian formulation).

In the next section, we will describe in more detail how we compose the datasets and elaborate on our experiments.

## 3. Results

All 3D models, datasets, plots and trained GP models described below can be found in our GitHub repository. These include supplementary experiments and plots we omitted here, so we do not overload the text.

To assess the latent representation of various 3D shapes, we composed a collection of both synthetically generated point clouds and real-world scanned objects. An overview can be found in Table 1. The synthetically generated point clouds are based on objects drawn in the free and open source Blender 3.3 LTS (https://www.blender.org/, accessed on 10 January 2023). Point clouds resulting from real-world scans were created on an Android mobile phone using the photogrammetry KIRI engine (https://www.kiriengine.com/, accessed on 10 January 2023) and imported in Blender, where they are cleaned up by dissolving disconnected points, removing the background and subsampling using standard Blender tools. However, they still contain some overlapping triangles and other mesh

irregularities. Synthetic models were made noisy by first subdividing the mesh several times and then applying the Blender Randomize tool on the vertices. This breaks the lattice structure of the vertices. Moreover, this makes them resemble a real-world scan, where imperfections are inevitable. In this paper, we restrict ourselves to one noise level and leave the effect of the amount of noise on our point clouds as future work. The noiseless and noisy bend torus are the same as their ordinary torus variant with a Simple Deform modifier of 45° applied to it. All models are exported in the Polygon File Format format (.ply), resulting in files consisting of points and unit normals for those points.

The line elements are calculated in Matlab R2020b and exported as comma-separated values (.csv). These serve as the data space for the GPLVM models, which are implemented using the python GPy library (http://sheffieldml.github.io/GPy/, accessed on 10 January 2023). Some point clouds were subsampled uniformly for performance reasons. All GPLVM models were initialised with the results from a PCA. The BCGPLVM models were implemented with an MLP mapping with five hidden layers. The details are in Table 1. All code for training the models, as well as the trained models themselves, are available via notebooks in the GitHub repository.

**Table 1.** An overview of the surfaces and their corresponding GPLVM and properties. Larger point clouds are subsampled uniformly to a smaller set. The number of points retained for training is given in # Train. Noise is the Gaussian noise variance hyperparameter for the GPLVM model, which is either a fixed value or a value that has to be learned along with the other hyperparameters. The number of inducing points for the Bayesian GPLVM is shown in # Ind. To make sure we do not end up in local minima, we restarted the training of the model a number of times given in # Restarts.

| | Model | # Vertices | # Train | Noise | # Ind | # Restarts |
|---|---|---|---|---|---|---|
| Cylinder of revolution | BGPLVM | 2176 | 2176 | $1 \times 10^{-4}$ | 15 | 10 |
| Cone of revolution | BGPLVM | 2176 | 2176 | free | 50 | 10 |
| Spiral cylinder | BGPLVM | 2210 | 2210 | free | 25 | 10 |
| Cylinder w/o revolution | BGPLVM | 1717 | 1717 | free | 50 | 10 |
| Cone w/o revolution | BGPLVM | 2210 | 2210 | free | 50 | 10 |
| Surface of revolution | BGPLVM | 2816 | 2500 | free | 50 | 10 |
| Helical surface | BGPLVM | 2582 | 2500 | free | 25 | 10 |
| Spiral surface | BGPLVM | 1842 | 1842 | free | 50 | 10 |
| Torus | BGPLVM | 2048 | 2048 | free | 50 | 10 |
| Torus bend | BGPLVM | 2048 | 2048 | free | 25 | 10 |
| Pear | BGPLVM | 6356 | 5000 | free | 50 | 5 |
| Mixture 1 | BCGPLVM | 10,653 | 1000 | $1 \times 10^{-6}$ | NA | 3 |
| Mixture 2 | BCGPLVM | 6555 | 1000 | $1 \times 10^{-6}$ | NA | 3 |
| Mixture 3 | BCGPLVM | 15,681 | 1000 | $1 \times 10^{-4}$ | NA | 3 |
| Hinge | BGPLVM | 22,282 | 5000 | free | 50 | 3 |
| Torus bend noisy | BGPLVM | 8192 | 8192 | free | 50 | 3 |

*3.1. Surface Approximation*

We trained a Bayesian Gaussian process latent variable model for all examples of the equiform kinematic surfaces listed in Section 2.2. As can be seen by the ARD contributions in Figure 2, these surfaces can be represented by a lower dimensional representation. These dimensions are characterised by the largest ARD contribution and, thus, the smallest length scales. A 2D plot of the surface points in their latent space is given by Figure 3. The Supplementary Material also includes plots in 3D latent space and provides PCA bar charts for comparison. They are omitted here not to overload this text with too many plots. These experiments were repeated for the noisy variants of the surfaces as well. The same underlying structure can be observed. Again, we refer to the Supplementary Material.

All surfaces show a clear structure in their latent space. Note that the number of small eigenvalues does not necessarily correspond with the number of relevant dimensions in latent space. The latter is the result of an optimization algorithm in which both the latent points and the kernel hyperparameters are found. This can be seen in the ARD contribution

plot for the helical surface. The plot for the cylinder of revolution even has a significant value for all seven dimensions. This effect can be thought of as overfitting [37], as the model attributes importance to more latent dimensions than needed. In our experiments, we tried to lower this effect by making the model less flexible. We added a fixed noise term to the hyperparameters and lowered the number of inducing points. For details per surface, we refer to Table 1. All trained models are available in the Supplementary Material on the GitHub repository.

Another important remark is that the mapping from latent to data space is non-linear. Care must be taken when interpreting the 2D latent space plots. For instance, the spiral surface clearly has a one-dimensional subspace. However, the 2D plot shows a scattered cloud of points. This is an artefact of the visualization. The ARD plot indicates that only dimension 1 has a significant contribution. Another example is given by the cylinder without revolution. Its subspace in $\mathbb{R}^7$ is one-dimensional, which manifests itself as a curve-like trail of points in the 2D latent space.

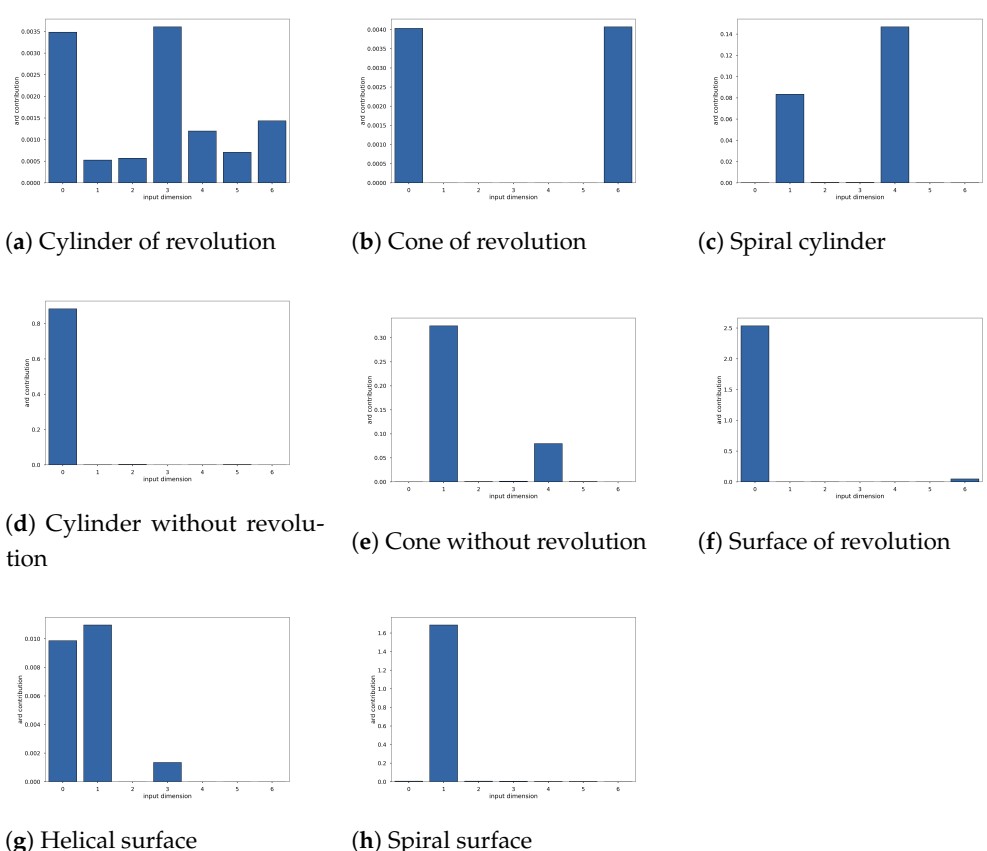

(**a**) Cylinder of revolution

(**b**) Cone of revolution

(**c**) Spiral cylinder

(**d**) Cylinder without revolution

(**e**) Cone without revolution

(**f**) Surface of revolution

(**g**) Helical surface

(**h**) Spiral surface

**Figure 2.** ARD contributions for the dimensions of the latent space for the examples of equiform kinematic surfaces.

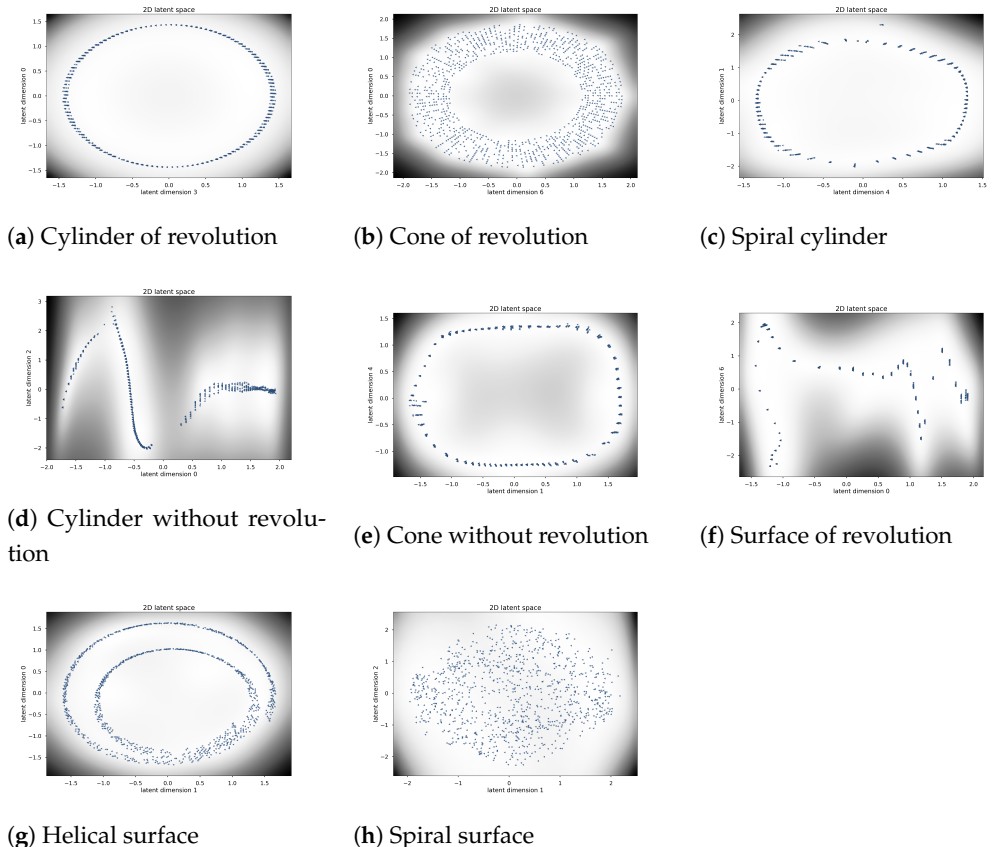

**(a)** Cylinder of revolution      **(b)** Cone of revolution      **(c)** Spiral cylinder

**(d)** Cylinder without revolution      **(e)** Cone without revolution      **(f)** Surface of revolution

**(g)** Helical surface      **(h)** Spiral surface

**Figure 3.** A 2D representation of the points of the kinematic surfaces in their latent space. The amount of black in the background indicates the posterior uncertainty of the BGPLVM.

So far, nothing has been gained by this new Bayesian GPLVM way of representing surfaces. The difference with the approach described in Section 2.2, is that we are no longer restricted to the simple geometric surfaces of Figure 1 and their linear subspaces of $\mathbb{R}^7$. We can now also describe surfaces that do not fall into the categories listed above. We investigate two cases.

First, we apply our method to a bend torus. This is a surface of revolution, which we altered using the Simple Deform modifier in Blender to bend it 45° around an axis perpendicular to the axis of rotational symmetry of the original torus. This removes the rotational symmetry altogether. The results can be seen in Figure 4. We notice that the BGPLVM only deemed one dimension as significant. The 2D plot reveals the latent structure.

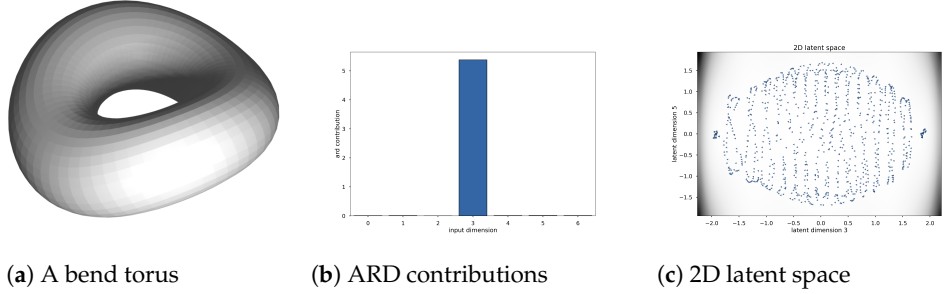

**(a)** A bend torus      **(b)** ARD contributions      **(c)** 2D latent space

**Figure 4.** Results for a bend torus. One latent dimension is found to be dominant.

Second, we look at the surface of the point cloud obtained by scanning a pear as described above. This is an organic shape, so it possesses the same challenges as working with shapes that can be found in other items from nature or when modelling the anatomy of humans and animals. We are only interested in the shape of the body, so we removed

the stalk and the bottom part when cleaning up the 3D model. In this case, the 3D shape resembles a surface of revolution, but the axis is bent irregularly and the rotational symmetry is broken (not all normals intersect the axis of rotation). The results can be seen in Figure 5. The darker region in the 2D latent plot indicates more posterior uncertainty. In the latent space, we observe a set of points similar to what we saw for a cylinder of revolution with an additional distortion in a third latent dimension.

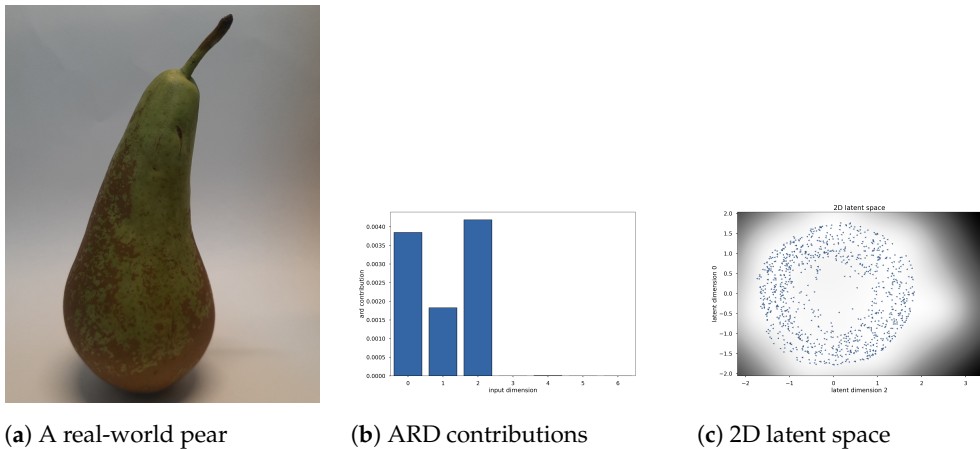

(**a**) A real-world pear      (**b**) ARD contributions      (**c**) 2D latent space

**Figure 5.** Results for a real world pear scanned with a mobile phone app.

Both the bend torus and the pear can not be described by an equiform kinematic surface. Applying the methods of Section 2.2 (i.e., approximating by a linear complex of line elements) for classification is numerically still possible. A small set of eigenvalues can be found. However, their interpretation would be faulty. The bend torus shows one small eigenvalue, $\mathbf{c} \neq (0,0,0)$, $\gamma = 0$ and $\mathbf{c} \cdot \bar{\mathbf{c}} \approx 0$. Unsurprisingly, these values fit a surface of revolution. They resemble the values for the torus or the torus with noise. However, blindly using the methods from [10] would result in a perfect surface of revolution. The same reasoning can be applied to the scanned pear's point cloud. Below, we show how to exploit our newfound GPLVM representation in surface approximation, surface segmentation and surface denoising.

### 3.2. Surface Segmentation

A major challenge in point cloud classification is the segmentation of sub-regions within that cloud. Once points are grouped together in simpler shapes, the underlying structure can be found via either our method or the methods described in [8,10–13,25,26]. In these works, several approaches are described for discovering the sub-regions. Mostly, they are based on time-consuming trial and error RANSAC. Here, we show that working in a latent space can be beneficial. The challenge is to group points together, whose line elements show similar behaviour.

As we want to separate coherent groups of points in latent space, we care about their local distances. Points close by in the latent space should be close by in the data space as well. Therefore, we expand our GPLVM with a back constraint as described in Section 2.5. We implement both an RBF kernel with ARD and a multi-layer perceptron (MLP) mapping to capture the back constraint [33,37]. The details for the different 3D models can be found in Table 1. As before, all code is available in the GitHub repository. The notebooks also include 3D plots made with the python open source graphing library Plotly (https://plotly.com/python/, accessed on 10 January 2023), that allow user interaction such as 3D rotations. By rotating the viewpoint, we can clearly see how separable the latent points are.

To demonstrate our approach, we first designed three objects composed of different simpler geometric shapes. They can be found in Figure 6. The parts of these three models

fall under the different categories described in Section 2.2. The aim of surface segmentation is to find those parts in an unsupervised manner.

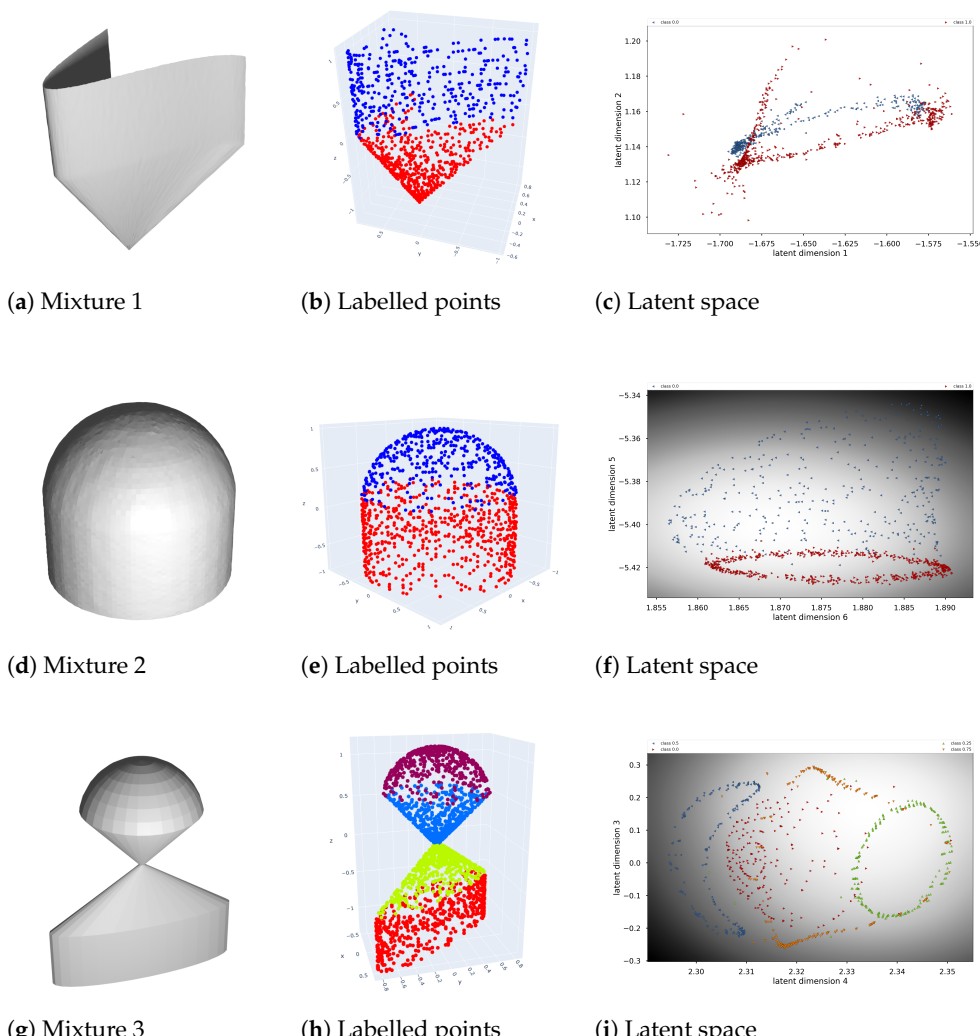

(**a**) Mixture 1    (**b**) Labelled points    (**c**) Latent space

(**d**) Mixture 2    (**e**) Labelled points    (**f**) Latent space

(**g**) Mixture 3    (**h**) Labelled points    (**i**) Latent space

**Figure 6.** Three synthetically generated 3D models by combining primitive surfaces. The BCGPLVM is able to show distinguishable structures for points in latent space.

First, we created a 3D model called Mixture 1, which consists of a cylinder and cone, neither without rotational symmetry. Both of those shapes individually show one small eigenvalue and a clearly distinguishable curve in their 2D latent space, as can be seen in Figure 3. Combined, their latent space looks like two curves, shown in Figure 6. Notice that the 3D points that lie both on the cylinder and the cone, fall into both categories. Moreover, their normal is inconsistent with either of the two shapes. For this cylinder, all normals are horizontal. For the cone, normals for points on a line connecting the cone's apex and its generating curve, are parallel. For points on the intersection of the cylinder and the cone, the normals are weighted with their neighbouring points. This results in a latent space that is not easily separable by clustering.

Second, the 3D model named Mixture 2 consists of a noisy cylinder of revolution where one end is closed by a demi-sphere. The former is characterised by two small eigenvalues and the latter by three. Again, this behaviour can be clearly observed in the latent space. Notice how the BCGPLVM formulates latent shapes for each part that are consistent with the kinematic surface described in Section 2.2. For the sphere, we observe a 2D shape. For the cylinder of revolution, an annulus can be seen. The supplementary material includes an interactively rotatable 3D plot where this cloud of latent points can

be observed in more detail. We also see that the region for the tip of the demi-sphere has a darker background in the 2D latent plot, indicating more uncertainty in this region of the posterior. This can be explained by the fact that the normals of a sphere all intersect at the centre of the sphere. As such, no normals are parallel. This results in line elements whose vector components vary more. Points with normals that lie in parallel planes, as is the case for a cylinder, have more similarity in the direction components of their line elements. Moreover, the hyperparameters in the mapping from latent to data space are optimised globally. This means for all latent points simultaneously. The strong structure in the cylinder part renders the large variations in the tip of the demi-sphere part as less likely. Hence the larger posterior variance.

Finally, in the 3D model Mixture 3, we grouped together the upper half of a sphere, a cone of revolution, a cone without revolution and a cylinder without revolution. These parts have three, two, one and one small eigenvalues, respectively. As this model consists of four different parts, the segmentation is more complex. Nonetheless, the BCGLVM is able to find distinct substructures in the latent space, even in just two dimensions.

For a real-world and more challenging example, we scanned a metal hinge, as described above. It can be found in Figure 7. The original 3D model and the cleaned-up one can be found in the supplementary material. The 3D model is a collection of a cylinder of revolution, two planes and a cone-like aperture. It is important to notice that the scan itself is of poor quality, mainly due to the shininess of the metal and the lack of distinct features. There are holes and bumps in the surface, even after cleaning up the model in Blender. Moreover, the cone-like aperture does not have a lot of vertices (the region around the apex is completely missing). The latent space still shows the formation of clusters, especially when three dimensions are taken into account.

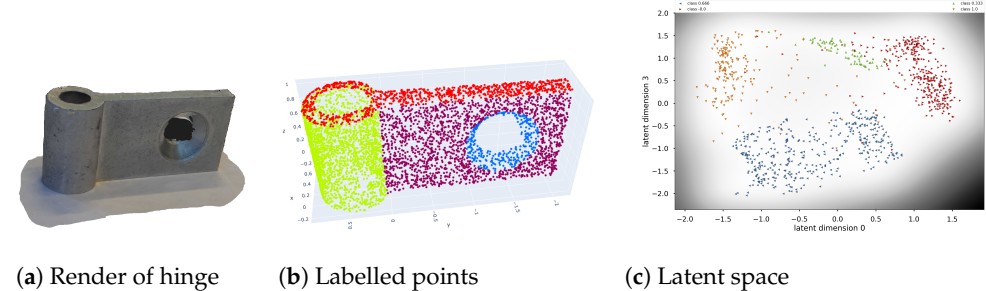

(**a**) Render of hinge     (**b**) Labelled points     (**c**) Latent space

**Figure 7.** The results for a real-world scanned metal hinge. Again, the BGPLVM is able to separate the points in latent space.

Once a latent space is found, the segmentation can be done via either a manual selection of the latent points or a form of unsupervised learning. In the case of separable clusters, we can perform the well-studied k-means clustering algorithm or draw (hyper)planes determined by support vector machines (SVM). The details of these are outside the scope of this work. The reader is referred to [38] and [39], respectively. This segmentation can then be the basis for fitting simple geometric surfaces to each cluster of points. As we can observe from the plots, some of the latent points do not belong to any of the found substructures. In practice, these can be ignored or filtered out. We are left with enough points to perform the best fit. Afterwards, we can determine whether or not such a rogue point belongs to the best-fit subsurface or not.

### 3.3. Surface Denoising

In general, a Gaussian process can handle noise very well, even in low data regimes [15]. This means our technique is beneficial to denoise point clouds. Once a mapping is found from a latent space to a data space, it can be queried to predict new points in the data space. This can be used to handle missing data [36,37]. Here, we take advantage of this feature by correcting noisy points in the lower dimensional latent space and predicting their

counterparts in the data space. The smooth mapping allows re-predict the line elements for every latent point.

From a predicted line element $(\mathbf{l}, \bar{\mathbf{l}}, \lambda)$, with $\|\mathbf{l}\| = 1$, we can calculate a corresponding 3D coordinate $\mathbf{x}$ for a point $x$ using

$$\mathbf{x} = \mathbf{l} \times \bar{\mathbf{l}} + \lambda \mathbf{l}. \tag{23}$$

To demonstrate this, we again work on the bend torus model. We introduce random noise with the Blender Randomize tool and select a hundred vertices at random, which we translate to simulate shot noise. The results can be seen in Figure 8. The 3D model, the .ply file with the point coordinates and unit normal vectors, the .csv file with the line elements and the notebook with the executed code for the BGPLVM can be found in the supplementary material. Once the BGPLVM is trained on the noisy point cloud, we use it to predict line elements for the latent point. From these line elements, we extract 3D coordinates for points via Equation (23). We observe that the BGPLVM is able to smooth out the translated vertices. This approach can also be used to detect and remove outliers.

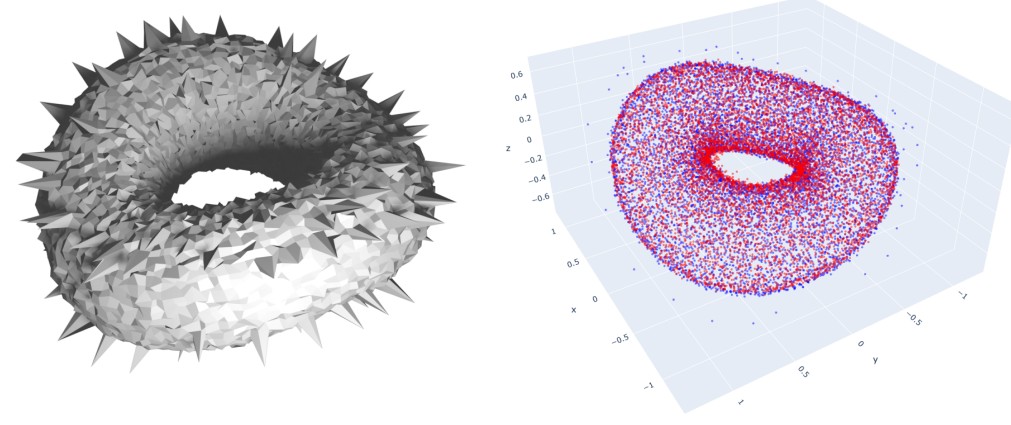

(**a**) Deformed bend torus　　　　　　(**b**) Denoised bend torus

**Figure 8.** A bend torus. Noise is added to the entire surface. Moreover, A hundred vertices were translated. (**b**) The BGPLVM is able to smooth out the surface. Blue are the noisy points. Red are the denoised points.

## 4. Discussion

This work presented the first findings for this new GPLVM approach to describe 3D surfaces. In this manuscript, we wanted to focus on the theoretical principles themselves and not overload the paper with additional research questions that determine the limits of this idea. Even though these are both interesting and important in real-world applications, we leave them for future work.

We have shown surface segmentation for surfaces that are the combination of a few different simpler geometrical shapes. The question remains how many sub-regions can be detected and what the complexity of those regions can be?

We presented the Bayesian GPLVM and the GPLVM with back constraints. There are more variations on this topic investigated in the literature. A recent paper describes a generalised GPLVM with stochastic variational inference [37]. They also present models for applying these techniques on a larger dataset. This would be most applicable to larger point clouds, which are often obtained in real-world applications.

A line element is formed by a line and a point on that line. By working with normal lines for points on a surface, we effectively introduced a second so-called *view* for those points, where we follow the terminology used in [34,36]. Those works present a multi-view unsupervised learning technique called manifold relevance determination (MRD), which offers another worthwhile approach.

The prediction as seven-tuples made by the model does not automatically follow the Grassmann–Plücker relationship in Equation (3) for their direction and moment vector. This leads to faulty line elements. In other words, the first six components of a line element vector are the Plücker coordinates of the line where the point of the line element lies on. Not all six-tuples represent a straight line in $\mathbb{P}^3$. In general, a *screw centre C* can be written as $(\mathbf{c}, \bar{\mathbf{c}})$. The pitch of $C$ is defined as

$$\rho = \frac{\mathbf{c} \cdot \bar{\mathbf{c}}}{\|\mathbf{c}\|^2}. \tag{24}$$

This only holds for $\mathbf{c}$ not being the zero vector, in which case $C$ would be a line at infinity. The pitch can be thought of as the deviation of the screw to a perfectly straight line. We can always write $C$ as

$$C = (\mathbf{c}, \bar{\mathbf{c}} - \rho\mathbf{c}) + (\mathbf{0}, \rho\mathbf{c}) = A + (\mathbf{0}, \rho\mathbf{c}), \tag{25}$$

in which $A$ is called the *Poinsot* or *central axis* of the screw centre $C$. The term $(\mathbf{0}, \rho\mathbf{c})$ represents the line at infinity where the planes perpendicular on $A$ meet. Since $A$ does satisfy the Grassmann–Plücker relation, it is a straight line in $\mathbb{P}^3$. This allows us to correct the predicted six tuples (by six distinct Gaussian processes) into straight lines via

$$A = C - (\mathbf{0}, \rho\mathbf{c}). \tag{26}$$

This approach is taken in [40]. Another way to ensure the Grassmann–Plücker relation is given in [41], where constraints are built in the kernel functions of the Gaussian process themselves, although at a considerable extra computational cost. A representation that does not suffer from this hurdle is the stereographic projection of a line [24]. In this approach, the line is made to intersect two arbitrary parallel planes. Only the 2D coordinates on those planes are kept as data points. Predictions on those planes, a point on each, are then used to calculate the Plücker coordinates of the predicted line. By doing so, the Grassmann–Plücker relation is always ensured. The problem herein is that the line parallel to the two planes can not be captured. Numerically, a line close to being parallel to the two planes also causes issues. As 3D surfaces can have normals in any direction, this latter approach is not recommended in a general setting.

## 5. Conclusions

We provided a theoretical introduction to kinematic surfaces and showed how they could be used to perform surface detection. Many simple geometric shapes manifest themselves as linear subspaces of line or line element space. This approach is limited by the linearity of the underlying eigenvalue problem. We expanded on this by reformulating this as a probabilistic non-linear non-parametric dimensionality reduction technique known as the Gaussian process latent variable model. We showed how this could be applied to many simple geometric surfaces, as well as surfaces that do not fall into any of these categories. Moreover, we showed the benefits of unsupervised surface segmentation and surface denoising. We presented findings on synthetically generated surfaces and scanned real-world objects.

The main goal of the current study was to determine the feasibility of applying the Gaussian process latent variable model to line element geometry. Even though several experiments are explained, and several more are included in the supplementary material, considerably more work will need to be done to determine the limits of this method. For instance, it remains an open question how noise affects the overall representation in the latent space. Moreover, we did not implement any optimizations on the training part of the underlying models, which is paramount for real-world settings. We leave this as future work.

Another natural progression of this work is to exploit further the found latent space in the case of missing data. Point clouds sometimes have missing regions, caused by bad lighting conditions, occluded areas or areas that simply can not be reached by the scanning device. Finding the 3D coordinates for the missing points is a classic example of the missing data problem. In our case, it manifests itself as a region in the latent space that is missing values. If the found structure in the latent space is enough to reconstruct those missing latent points, then according to data space points can also be inferred by the Gaussian process latent variable model.

More broadly, we plan to study the benefits of working on latent spaces not just for line elements, but for the lines themselves (in which case, we drop the point on the line and only keep the description of the line). This technique could be used in the calibration of various devices that are built on the usage of straight lines. Cameras, for instance, produce images in which a pixel is the result of a single incoming ray of light. Another example is galvanometric laser scanners, which guide a laser beam by means of two rotating mirrors. Calibrating such a device means finding the relationship between the two angles of the mirrors and the outgoing beam. So, in this case, a 2D latent space must exist. This would be a fruitful area for further work.

**Supplementary Materials:** All of our 3D models, sets of line elements, trained GPLVM, notebooks with code and many more experiments and plots can be found on our GitHub repository https://github.com/IvanDeBoi/Surface-Approximation-GPLVM-Line-Geometry (accessed on 10 January 2023).

**Author Contributions:** Conceptualisation, I.D.B., C.H.E. and R.P.; methodology, I.D.B., C.H.E. and R.P.; software, I.D.B.; validation, I.D.B.; formal analysis, I.D.B. and R.P.; investigation, I.D.B.; resources, I.D.B.; data curation, I.D.B.; writing—original draft preparation, I.D.B.; writing—review and editing, I.D.B., C.H.E. and R.P.; visualisation, I.D.B.; supervision, R.P.; project administration, I.D.B.; funding acquisition, R.P. All authors have read and agreed to the published version of the manuscript.

**Funding:** This research has been funded by the University of Antwerp (BOF FFB200259, Antigoon ID 42339).

**Institutional Review Board Statement:** Not applicable.

**Informed Consent Statement:** Not applicable.

**Data Availability Statement:** All of our 3D models, sets of line elements, trained GPLVM, notebooks with code and many more experiments and plots can be found on Supplementary Materials.

**Acknowledgments:** We thank the anonymous reviewers whose comments helped improve the quality of this paper.

**Conflicts of Interest:** The authors declare no conflict of interest.

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
