# Peer review of "Surface Approximation by Means of Gaussian Process Latent Variable Models and Line Element Geometry"

_mathematics, doi:10.3390/math11020380_

Round 1

Reviewer 1 Report

Please find attached the review report. 

Reviewer 2 Report

This study introduces the GPLVM as an alternative to solve the surface detection and reconstruction tasks. The overall idea is straightforward. But what is lacking is the rigorous comparison between the proposed method and the existing methods. It seems that the authors just throw out this idea and preliminarily shows it works. It is hoped that the authors can show more examples to prove the advantage of their model from different perspectives (e.g., performance and accuracy).

Typo: line 189, "Forth" should be "Fourth".
